# The Immune Response in Adipocytes and Their Susceptibility to Infection: A Possible Relationship with Infectobesity

**DOI:** 10.3390/ijms23116154

**Published:** 2022-05-31

**Authors:** Orestes López-Ortega, Nidia Carolina Moreno-Corona, Victor Javier Cruz-Holguin, Luis Didier Garcia-Gonzalez, Addy Cecilia Helguera-Repetto, Mirza Romero-Valdovinos, Haruki Arevalo-Romero, Leticia Cedillo-Barron, Moisés León-Juárez

**Affiliations:** 1Université Paris Cité, INSERM UMR-S1151, CNRS UMR-S8253, Institut Necker Enfants Malades, 75015 Paris, France; olopezortega@outlook.com; 2Laboratory of Human Lymphohematopoiesis, Imagine Institute, INSERM UMR 1163, Université de Paris, 75015 Paris, France; moreno.corona.n@gmail.com; 3Departamento de Immunobioquímica, Instituto Nacional de Perinatología Isidro Espinosa de los Reyes, Ciudad de México 11000, Mexico; vic_cruise@hotmail.com (V.J.C.-H.); biol.didier@gmail.com (L.D.G.-G.); ceciliahelguera@yahoo.com.mx (A.C.H.-R.); 4Departamento de Biomedicina Molecular, Centro de Investigación y de Estudios Avanzados del Instituto Politécnico Nacional (CINVESTAV-IPN), Av. Instituto Politécnico Nacional 2508, San Pedro Zacatenco, Mexico City 07360, Mexico; lcedillo@cinvestav.mx; 5Departamento de Biología Molecular e Histocompatibilidad, Hospital General “Dr. Manuel Gea González”, Calzada de Tlalpan 4800, Col. Sección XVI, Ciudad de México 14080, Mexico; mirzagrv@yahoo.com; 6Laboratorio de Inmunología y Microbiología Molecular, División Académica Multidisciplinaria de Jalpa de Méndez, Jalpa de Méndez 86205, Mexico; haruki.arevalo@ujat.com

**Keywords:** obesity, infection, immune response, adipocytes

## Abstract

The current obesity pandemic has been expanding in both developing and developed countries. This suggests that the factors contributing to this condition need to be reconsidered since some new factors are arising as etiological causes of this disease. Moreover, recent clinical and experimental findings have shown an association between the progress of obesity and some infections, and the functions of adipose tissues, which involve cell metabolism and adipokine release, among others. Furthermore, it has recently been reported that adipocytes could either be reservoirs for these pathogens or play an active role in this process. In addition, there is abundant evidence indicating that during obesity, the immune system is exacerbated, suggesting an increased susceptibility of the patient to the development of several forms of illness or death. Thus, there could be a relationship between infection as a trigger for an increase in adipose cells and the impact on the metabolism that contributes to the development of obesity. In this review, we describe the findings concerning the role of adipose tissue as a mediator in the immune response as well as the possible role of adipocytes as infection targets, with both roles constituting a possible cause of obesity.

## 1. Introduction

We are currently facing an obesity epidemic, as obesity is becoming a growing health problem worldwide and has a complex and multifactorial etiology [1]. According to the World Health Organization (WHO), obesity and overweight are defined as excessive fat accumulation that leads to higher morbidity rates for various health problems [2].

Data from several databases focused on the obesity research field indicate that susceptibility to infections and obesity are related. These databases have also identified adipocytes as key components in the crosstalk between microorganisms, obesity, and immune cells [3], thus opening up a new landscape in this research field and prompting the emergence of a new “infectobesity theory”. This theory is sustained by multiple lines of evidence: Firstly, many reports have described how infections by pathogens alter the architecture of adipose tissue due to inflammation or pathogen persistence [4]. Additionally, there is evidence that supports the role of infected adipocytes in obesity, since some pathogens are able to dysregulate components (i.e., signaling pathways) of adipose tissues and participate in the development of obesity [1,4]. This theory maintains that some microorganisms are capable of inducing the obesogenic process [5,6], with adenovirus 36 being one of the most studied pathogens associated with obesity.

Adipose tissue is a specialized connective tissue emerging as a critical metabolic organ that regulates energy homeostasis in the body [7,8]. Its main function is to store energy in the form of neutral lipids (fat) [8]. However, growing evidence now indicates that adipose tissue can sense and integrate mixed signals, such as signals of a hormonal, metabolic, inflammatory, and even neuronal origin [9,10,11]. Adipose tissue consists of lipid-rich cells called adipocytes and can be divided into white, pink, beige, and brown adipose tissue. White adipose tissue is crucial in the host’s metabolism due to its capacity to store and release fatty acids, which supply fuel to the organism during fasting periods; on the other hand, brown adipose tissue helps to transform energy into heat to regulate body temperature by thermogenesis [7,8]. Nevertheless, excessive adipose tissue is associated with several chronic diseases [12].

Adipocytes have been related to different pathologies such as diabetes and metabolic syndrome [13,14]. Likewise, adipocytes have recently been described as natural targets of several infections [15] that trigger the alteration of the normal metabolic response. Moreover, obesity has been considered as a state of low-grade inflammation due to an excess of cytokines and adipokines, which modify glucose and insulin metabolism [12] and a combination thereof, thus leading to insulin resistance and metabolic syndrome.

A comprehensive understanding of the relationship between adipocytes, cellular metabolism, infections, and immune responses may further bring insight into how the specific responses to pathogens can act as etiological factors in obesity. In this review, we will focus on how adipocytes’ immunological mediators can respond to infections whilst also contributing to obesity.

## 2. The Role of Adipose Tissue in the Immune Response

### 2.1. Adipose Tissue’s Composition

Adipose tissue is a dynamic organ distributed throughout the body [16]. For many years, adipose tissue (AT) was considered an inert mass; however, the recognition of its importance has been increasing [8]. Adipose tissue is currently regarded as a critical endocrine system (organ) that releases cytokines, chemokines, and adipokines, including adiponectin, leptin, Tumor Necrosis Factor-α (TNF-α), Interleukin (IL)-1β, IL-6, and others that are crucial in the regulation of energy and immune responses [8].

Adipose tissue is organized into discrete depots throughout the body [17,18]. Brown and beige adipocytes represent a small proportion of the total adipose tissue [18], with white adipose tissue (WAT) being the most abundant form as it is found in almost every area of the body. In fact, white adipocytes change their phenotype during pregnancy and lactation (transdifferentiation), developing into pink adipocytes [19,20]. The major WAT depots can be broadly classified by location, as either subcutaneous (under the skin) or visceral and as visceral/omental (located intra-abdominally, adjacent to internal organs) [17,18].

Although adipocytes account for most of the volume of AT, they only make up about 50% of the cellular content [16]. Other cell types, including endothelial cells, fibroblasts, and immune cells, are present in this tissue [17]. Macrophages and T lymphocytes are the principal immune cells infiltrating adipose tissue, and one of their functions is to secrete proinflammatory mediators [21,22]. In more detail, visceral AT, in contrast to subcutaneous AT, tends to have a higher content of macrophages, regulatory T cells, natural-killer T cells, and eosinophils [16], suggesting that it could be a potential immunologic effector site; however, more studies need to be conducted to position adipose tissue as a new immune site.

### 2.2. Adipokines

Adipokines are a family of hormones and cytokines that are secreted by adipose tissue and can modulate diverse cellular responses in immune cells, such as cell migration [23] or even the cellular subpopulations of T cells [24]. Among these proteins, leptin and adiponectin are the principal regulators of the adipocytes, and it is for this reason that we will focus on them [25].

#### 2.2.1. Leptin

Leptin is secreted by adipocytes in response to changes in a person’s nutritional status. Leptin is a 167-amino-acid product of the human leptin gene. It was identified for the first time via a defect in the gene responsible for the obesity syndrome in leptin-deficient (ob/ob) mice [26,27]. Leptin is mainly secreted by white adipose tissue, and its levels are positively correlated with the amount of body fat [27].

The main function of leptin is to modulate energy homeostasis. To achieve this, its signals are focused on the central nervous system (CNS) and peripheral organs, promoting satiety and decreasing food intake and metabolism. However, leptin can also act peripherally to increase glucose utilization by skeletal muscles and to decrease gluconeogenesis in the liver. The most significant role of leptin is its neuroendocrine function, mediated through pathways that overlap with insulin, as recently shown in both animal models and humans [25,26,28].

Leptin mediates its effects by binding to specific leptin receptors (ObRs) expressed in the brain and peripheral tissues [27]. The binding of leptin to the ObR receptor activates several signal transduction pathways, including Janus Kinase (JAK)-Signal Transducer and Activator of Transcription 3 (STAT3), as well as Phosphatidylinositol 3 Kinase (PI3K) and Mitogen-Activated Protein Kinases (MAPK) [25,27].

Leptin also activates the AMP-activated Protein Kinase (AMPK) pathway, promotes fatty acid oxidation by inactivating acetyl-coenzyme A (CoA) carboxylase, and inhibits lipogenesis. In mouse models, lipogenic transcription factors, cholesterol biosynthesis, plasma, and tissue lipids are also regulated by leptin [28].

In addition to its regulatory function in the CNS, leptin is a proinflammatory adipokine that plays a major role in innate and adaptive immunity. Leptin protects T lymphocytes from apoptosis and regulates their proliferation and activation. Leptin also influences the cytokine production of T lymphocytes, generally by switching the phenotype toward a TH1 response [29,30,31].

In monocytes, leptin also influences activation, phagocytosis, and cytokine production (TNF-α, IL-1, IL-6, IL-8, and IL-18) and cell proliferation. In turn, leptin expression is also elevated by proinflammatory cytokines such as TNF-α and IL-1, indicating a bidirectional interaction between leptin and inflammation [25,26,27,32].

#### 2.2.2. Adiponectin

Adiponectin is the most abundant adipokine secreted by adipocytes. It circulates at the highest levels (in the microgram per milliliter range versus the nanogram per milliliter range for leptin). This adipokine is best known for its role in the regulation of insulin sensitivity [25].

The central functions of adiponectin are orchestrated via AMPK signaling. In subjects with increased body masses, the downregulation of adiponectin has been detected, and adiponectin levels have been shown to be inversely correlated with glucose intolerance and type 2 diabetes. In this context, adiponectin appears to promote beta-cell function and survival [32]. Moreover, adiponectin enhances insulin secretion by stimulating both the expression of the insulin gene and the exocytosis of insulin granules [33]. In general, obese individuals release less adiponectin than non-obese people [34]. Adiponectin increases insulin sensitivity by stimulating insulin receptors via tyrosine phosphorylation and by the activation of AMPK, which leads to fatty acid oxidation [35]. Additionally, adiponectin can decrease (Reactive Oxygen Species) ROS production and inhibit the activation of MAPK, leading to the inhibition of cell proliferation [36].

Adiponectin presents specific anti-inflammatory properties through the inhibition of IL-6 production, through the induction of anti-inflammatory cytokines such as IL-10 or an IL-1-receptor antagonist, and through a reduction in ICAM-1 and VCAM-1. In fact, older ICAM-1-deficient mice, or even those lacking CD11b/CD18 (one of the main ligands for ICAM-1), have shown an increase in obesity [37]. Additionally, adiponectin levels are decreased in obese and type 2 diabetic subjects and negatively correlate with visceral mass [32].

### 2.3. Adipocyte Mass and Function and Their Relationship with the Immune Cells

From an immunologic point of view, obesity is considered as a state of low-grade inflammation of adipose tissue, which leads to the chronic activation of the immune system that alters diverse cell processes such as insulin resistance [12] and even diabetes. Additionally, obesity is associated with alterations in the function and number of immune cells; for example, increases in phagocytosis [38] and oxidative bursts [39] in monocytes and granulocytes and an increase in the number of leukocyte cells have been detected [40] (Figure 1). Studies on germ-free mice showed that these mice had 40% lower fat mass than mice in conventional conditions [41,42], suggesting that signals arising from the microbiome and immune response are vital for adipose tissue architecture and homeostasis.

The immune cells present in AT can also be affected by obesity; for example, adipose tissue macrophages from obese animals have more cells expressing I-Ab (mouse class II MHC molecule) that are implicated in antigen presentation as well as in the induction of T-cell proliferation [43], which is indicative of their essential role in immune regulation. Additionally, a study conducted in 2019 demonstrated that a high-fat diet (HFD) induced adiponectin expression. This led to a reduction in Interferon (IFN)-γ and IL-17-positive CD4+ T cells and dampened the differentiation of naïve T cells into Th1 cells and Th17 cells. They concluded that adiponectin reduces Th17 cell differentiation and restrains glycolysis in an AMPK-dependent fashion [24].

Interestingly, in obese subjects, the increased activity of Nuclear Factor Kappa B (NFkB) [44,45], a master regulator of proinflammatory cytokines [46], has been shown to lead to a decreased level of the NFkB inhibitor, IκBα [47]. This nuclear factor modulates the expression of IL-6, TNF-α [46], and MIP [48]; these cytokines are closely associated with obesity [49,50] and pathogen responses [51,52,53]. In fact, in obese subjects, the elevation of pro-inflammatory cytokines such as TNF-α or IL-6 has been noted [49,54]. Remarkably, the IL-6 secreted by adipocytes could reach up to 30% of the total IL-6 found in serum [55,56], suggesting that this tissue plays a vital role in modulating the immune response. The relationship between inflammation and obesity has been controversial. On the one hand, several studies have mentioned that inflammatory markers may be associated with weight gain, especially in middle-aged adults [57]. By contrast, some reports have noted a negative correlation between cytokine levels (IL-6, adiponectin, and TNF-α) and obesity [58], suggesting that inflammation is a consequence of obesity rather than a contributing factor [56].

Other signaling pathways related to obesity, infection, and immunity are MAPKs (especially C-Jun N-terminal Kinases (JNK)) and JAK–STAT [59,60,61]. Although JNK is one of the most addressed signal transducers in metabolic processes such as obesity and insulin resistance [62], JNK is mainly activated by growth factors and cytokines [63]. Additionally, in several tissues of obese subjects, inflammatory signals that activate JNK have been observed. The functional consequences of this include an increase in NFkB and Activator Protein 1 (AP-1) signaling, leading to more inflammatory signals (IL-1β, TNF-α, and IL-6) [64] that alter insulin sensitivity and later culminate in aggravated insulin resistance. Moreover, these proinflammatory signals modulate the Peroxisome Proliferator-Activated Receptor Gamma (PPAR-γ), controlling adiponectin expression and, later, the proinflammatory status [64].

JNK can modulate pathways other than NFkB, AP-1, and PPAR-γ. It can also activate the PTP1b (Protein-Tyrosine Phosphatase 1B) signaling pathway to regulate the insulin pathway in diverse cells and tissues [64]. For example, the expression and activation levels of PTP1b are increased in obese subjects, blocking insulin pathways. In contrast, PTP1b knockout mice show resistance to weight gain and have an increased insulin sensitivity even with a high-fat diet [64].

The importance of JNK in insulin resistance has been addressed in different ways; for example, with gene knockout technology, the deletion of JNK1 in adipocytes has been found to inhibit insulin resistance [65], whereas the lack of JNK2 was found to “boost” the sensitivity to insulin at a cellular level [66]. JNK signaling pathways are divergent; first, the lack of JNK1 leads to a decrease in the phosphorylation of IRS1 (Insulin Receptor Substrate 1), enhancing the insulin signaling pathway, which leads to a rise in cell sensitivity to insulin. Second, the absence or inhibition of JNK2 suppresses the expression and function of NFkB and AP-1, which lessens insulin resistance [67].

PPAR-γ is a key factor in adipocyte differentiation, lipid metabolism, atherosclerosis, insulin resistance, and inflammatory response [29]. The activation of this molecule inhibits the expression of inflammatory-related genes such as those encoding Nitric Oxide Synthase (NOS), metalloproteinases, ILs, and cytokines in general [29], which contribute to insulin resistance.

It has been noted that TNF-α can block the transcriptional activity of PPAR-γ via Extracellular signal-Regulated Kinase (ERK) and JNK pathways [30]. This PPAR-γ downregulation impacts the expression of adiponectin, decreasing its levels and altering the modulation of insulin sensitivity in target cells. For example, fat cells that do not express PPAR-γ present lower levels of adiponectin, a reduced ability to store fat as well as decreased insulin sensitivity [68].

Finally, the JAK–STAT and MAPK signaling pathways are also altered in obesity [59,60,61]. The leptin receptor has been shown to activate JAK–STAT signaling pathways, mainly JAK2–STAT3, inducing the transcription of target genes [31,69]. It is worth mentioning that JAK2 can phosphorylate IRS1 and subsequently attract PI3K, thus activating downstream pathways [69]. On the other hand, JAK2 could also trigger the MAPK pathway, especially ERK1/2, in response to insulin, leading to the expression of several target genes such as c-fos and egr-1 [70], which participate in cell proliferation and differentiation [71]. In fact, the cardiomyocytes of C57bl6 mice stimulated with leptin showed the phosphorylation of STAT3 after 15 min of treatment [72]. Additionally, P38 MAPK is phosphorylated in response to leptin [59,72].

In general, diverse signaling pathways are activated by either microorganisms or immune signals that affect adipocyte homeostasis; this convergence highlights the infectobesity hypothesis as a new research field.

### 2.4. IFN Response

As previously mentioned, one of the most altered signaling pathways is JAK–STAT, with one of the most important downstream effectors being IFNs. Type I and type II IFNs are molecules formed in response to viral infections or cancer invasion and are fundamental to the initiation and regulation of the immune response. In general, IFNs directly induce antiviral responses within infected cells and are more critical for surrounding cells through the upregulation of molecules that can antagonize virus replication (e.g., IFN-stimulated effector genes (ISGs) that limit virus replication and spreading [73]). Moreover, as previously mentioned, several viruses have been related to obesity. Thus, IFNs—especially antiviral IFNs—appear to be pivotal actors in regulating adipogenesis, lipid metabolism, and membrane composition [74]. These phenomena are achieved by the repression of fatty acid and cholesterol synthesis and by controlling upstream mediators of the AMPK and the Mechanistic Target of Rapamycin (mTOR) complex [74]. In more detail, mTOR regulates IFN expression, and IFN regulates mTOR effectors [75], which leads to a mutual relationship.

On the other hand, obesity can drive changes in immune response such as by producing an attenuated and prolonged IFN response that would later result in antiviral inefficacy. One of the most closely related molecules is leptin. Leptin could upregulate the Suppressor of Cytokine Signaling 3 (SOCS3) [76] (the expression levels of SOCS3 are higher in obese patients than in non-obese subjects) [77]; this change in SOCS3 expression impairs type I IFN response (via JAK–STAT) as well as other functions in various immune cells. For example, the IFN-α/β production and altered signaling in respiratory epithelial cells and macrophages in obese patients as compared to those in non-obese individuals [78] were found to cause an increase in inflammatory responses that resulted in infectobesity.

Additionally, long-chain fatty acids can induce type I IFN responses in hepatocytes and macrophages. At the same time, a high-fat diet stimulates type I IFN-regulated genes in hepatocytes [79]. One of these genes is Interferon-Response Factor 3 (IRF3), the expression of which is positively associated with insulin sensitivity and negatively associated with type 2 diabetes in human adipose tissue. In the preadipocytes of mice, a lack of IRF3 induces an increase in PPAR-γ [80] and its proadipogenic genes, resulting in high adipogenesis and increased effects on adipocyte functionality. Furthermore, IRF3^−/−^ mice develop obesity, insulin resistance, glucose intolerance, and finally, type 2 diabetes due to an increase in proinflammatory macrophages (M1) and the concomitant loss of IL-10 in adipose tissue [79].

Additionally, IFN-γ and IFN-β regulate ceramide metabolism [74]; the levels of ceramide are considered as a biomarker of insulin resistance and obesity. It has been demonstrated that the lack of IFN-γ in mice (C57BL6) modestly increases insulin sensitivity, thus decreasing adipocyte size and regulating cytokine release. It has been suggested that IFN-γ is a pivotal molecule in regulating inflammation and glucose [81]. In 2009, McGillicuddy et al. demonstrated that IFN-γ downregulates the insulin receptor, IRS-1, and Glucose Transporter 4 (GLUT4), in addition to causing the loss of adipocyte triglyceride storage and a reduction in PPAR-γ, adiponectin, perilipin, and fatty acid synthase, among proteins. Additionally, they noted that the treatment with IFN-γ affected the differentiation of preadipocytes into mature adipocytes [81]. Moreover, obese IFN-γ^−/−^ mice showed a reduced expression of inflammatory genes (TNF-α and MCP1) in adipose tissue, which led to a decrease in local inflammation and an improvement in glucose tolerance [82].

Type I IFN is closely related to obesity; for example, an IFN-β1 overexpression model exhibited an inhibition of body weight gain independently of food intake [83]. Moreover, IFNs enhance glucose uptake in diverse cell types (MEFs and human pDCs) and increase ATP production (TCA cycle), glycolysis, and oxidative phosphorylation in immune cells, cancer cell lines, and keratinocytes [74].

The dysregulation of the IFN–obesity axis may result in the potential increase in a host’s susceptibility to viral infections, such as co-infections and even opportunistic infections in overweight patients [74].

The tissue-specific response induced by IFN has been previously described; for example, the deletion of IFNαr1 in hepatocytes exacerbated the steatosis and inflammation generated by a methionine- and choline-deficient diet. Additionally, a lack of this gene in adipose tissue was found to worsen the metabolic dysregulation induced by a high-fat diet, as indicated by weight gain, insulin resistance, and impaired glucose tolerance. By contrast, its deletion in intestinal epithelial or myeloid cells does not affect susceptibility to metabolic disease [79].

## 3. Adipocytes as Infection Targets and Their Relationship with Infectobesity

As mentioned previously, adipose tissue plays a vital role in immunity and the inflammation mediated by infiltrated or resident immune cells or by humoral factors such as cytokines or adipokines. Furthermore, several reports have established that obesity is a state of prolonged inflammation. This leads to the hypothesis that chronic inflammation due to an infection (infectobesity) in adipose tissue can cause obesity. Infectobesity is a phenomenon that has been characterized in diverse viral infections, and those pathogens can infect the adipocytes. Interestingly, viruses are not the only pathogens that can infect adipose tissue components; in recent years, it has been described how different pathogens can infect and alter lipid metabolism in the cell.

Diverse signaling pathways in adipose tissue are affected in response to infection, with the PI3K–AKT–mTOR axis being one of the most altered. This signaling pathway is involved in the control of nutrients, cellular proliferation, growth, survival, and mobility [84,85]. In the case of infections, PI3K–AKT–mTOR is a target for the modulation of expression or the state of activation. For example, *Trypanosome cruzi* (*T. cruzi*) infection increases PI3K expression and AKT activation in 3T3-L1 adipocytes [86], linking *T. cruzi* infection with the insulin–IGF-1 receptor cascade, and later with insulin resistance.

Additionally, leptin is essential for negatively modulating the immune response against infection; for example, the absence of leptin in ob/ob mice leads to a higher formation of granulomas in mice infected with mycobacteria. Moreover, an increase in leptin receptors has been described in Human Immunodeficiency Virus (HIV) infections [26]. Additionally, active tuberculosis is correlated with weight loss, cachexia, and low serum concentrations of leptin, which in turn suppresses lymphocyte stimulation and the secretion of Th1 cytokines such as IL-2, IFN-γ, and TNF-α [26].

Therefore, the relationship between adipocyte infection and the development of obesity can be a relevant point, and it is necessary to gain a better understanding of adipocytes’ participation in the factors that trigger weight gain. Hence, the next section focuses on diverse pathogens that have demonstrated tropism toward adipocytes, some of which are arising as new candidates for adipocyte dysregulation.

### 3.1. Adenovirus-36

Adenoviruses are the best characterized pathogens that have been associated with the development of obesity. Clinical data, meta-analysis, and serologic studies support the idea that the serotype of adenovirus-36 (Ad-36) is the only one related to the property of triggering obesity in humans [87,88]. The main research models to understand the mechanisms utilized by Ad-36 to promote obesity have mainly been focused on animals infected with Ad-36 as compared with noninfected animals [89,90]. The biochemical markers in serum that are associated with an altered lipid profile and obesity (cholesterol, triglycerides, and glucose) are interesting because their detection during Ad-36 infection is decreased as compared to that in controls and even animals subjected to high-calorie diets. This suggests a process involving the metabolic regulation of these molecules in adipose tissues [91,92].

The idea that a respiratory virus has tropism toward adipose tissue is complex. However, experimental evidence in animals infected with Ad-36 showed that the virus could spread to peripheral organs, including the spleen, kidney, liver, and brain, but had tropism toward adipose tissues. A finding that strongly supports this tropism was the identification of viral DNA in human adipose tissue samples, thus suggesting the adipogenic potential of this pathogen [93]. In humans, serological studies have been contradictory in proposing a correlation between obesity and antibodies against Ad-36, whereas other studies have identified the null relationship between these two events [94,95,96].

The key to the adipogenic process induced by Ad-36 is the direct infection of adipose cells, which induces an increase in adipocyte cell size (hypertrophy) and number (hyperplasia) both in vitro and in vivo [97,98,99]. In addition, several studies have shown the differentiation and proliferation of adipocytes in human primary cells, human mesenchymal cells, and muscle cells. In this context, the cellular models widely used to study this adipogenesis are the mouse fibroblast 3T3-L1 preadipocytes, which are differentiated from adipocytes under the stimulus of insulin and dexamethasone [100,101]. Interestingly, Ad-36 triggers the differentiation of adipocytes without any additional stimulus by inducing the expression of adipogenesis-related genes. The upregulated genes observed were those involved in the increase in the intracellular levels of triglycerides (LPL), molecules such as glycerol-3-phosphate dehydrogenase, PPAR-γ, and CCAAT/binding enhancer proteins α (C/EBP-α) and β (C/EBP-β) [101]. In fact, PPAR-γ is considered the master regulator of the adipogenic process, and the increase in these cellular markers strongly suggests the modulation of the differentiation events in mature adipocytes after infection by Ad-36. Furthermore, the infection generates a reduction in the synthesis and secretion of leptin of up to 52%, the accumulation and de novo synthesis of fatty acids, and an increase in glucose uptake of up to 93%. Thus, the authors of [100,101] pointed out that these findings may contribute to the adipogenic effect mediated by Ad-36 infection.

The viral protein E4Orf1 has been identified as the principal protein responsible for affecting the molecular pathways that are involved in adipogenesis during Ad-36 infection. This protein possesses a PDZ domain-binding motif (PBM) in the carboxyl-terminal region [102]. This domain lets E4orf1 interact with several cellular proteins involved in regulating growth and metabolism. One of them is PI3K, which, via the stimulation of the insulin receptor, leads to the translocation of GLUT4 to the plasma membrane in adipocytes and thus to the uptake of glucose [103,104]. In addition, another study demonstrated that the E4ORF1 protein of Ad-36 is responsible for adipogenesis and the proliferation of adipocytes in 3T3-L1 cells and human adipocyte stem cells (hASCs), highlighting this protein for its ability to upregulate PI3K [105]. The molecular mechanism by which E4ORF1 activates PI3K forms a ternary complex with the Drosophila disc-large protein (Dlg1), recruits the kinase at the membrane, and promotes its activation. At later stages, PI3K activates AKT to promote an increase in cellular metabolism, survival, and even oncogenic transformation [106] (Figure 2A).

All this implies that Ad-36 infection leads to adipogenesis in mice; however, in humans, this hypothesis needs further research. Although the majority of the papers describe Ad-36 as the main actor in weight gain by prompting adipocyte hypertrophy, the complexity of this phenomenon based on the relationship of immune cells, adipocytes, and Ad-36 is not fully understood; a network involving these three components as a possible cause for obesity may exist.

### 3.2. Human Immunodeficiency Virus

Another pathogen that can infect adipocytes is the HIV. This infection is mainly characterized by massive CD4+ T cell depletion in the intestinal mucosa, which then affects blood and lymphoid CD4+ T cells, thus causing sustained systemic immune activation and inflammation [107]. In addition, people infected with HIV suffer from metabolic alterations (including dyslipidemia, insulin resistance, and lipodystrophy). The HIV lipodystrophy syndrome was initially considered an adverse reaction to the use of Antiretroviral drugs (ARVs). However, these pathologies have been observed in untreated patients, indicating that the virus directly impacts metabolism [108]. The metabolic disturbances associated with HIV lipodystrophy include insulin resistance, an elevation in Low-Density Lipoprotein (LDL)-C and triglyceride levels, and a decrease in High-Density Lipoprotein (HDL)-C levels [109].

Early approaches demonstrated how leptin levels were higher and adiponectin levels were lower in patients with HIV lipodystrophy syndrome, which was manifested by central fat accumulation and peripheral fat loss [109].

Subsequent evidence suggested that HIV affected the adipose tissue, and this led to the theory that adipose tissue may constitute a viral reservoir. The first approach was carried out in 2002 using lysates of whole adipose tissue samples from twenty-three antiretroviral-treated HIV patients. However, only two patients were found to be positive for HIV RNA [110]. Afterward, in 2015, using more suitable techniques, HIV was found in 11 samples from adipose tissue of HIV-infected patients, and HIV RNA was detected in the stromal vascular fraction (SVF) of the AT using ultrasensitive PCR [107]. In a different study, HIV RNA was detectable in the AT-SVF cells from other adipose depots (visceral, subcutaneous, or deep neck) of all five HIV patients studied [111].

In this context, it was investigated whether adipocytes were permissive to HIV, or whether HIV affected adipose tissue metabolism. As mentioned previously, some lipodystrophy is present in these patients. Damouche et al. found that the simian immunodeficiency virus (SIV), which infects macaques, could lead to elevated immune activation and inflammation of adipose tissue. Both residents’ CD4+ T cells and macrophages were infected, and this led to further investigation into HIV-infected patients. The results showed the replication of an HIV component in ex vivo CD4+ T cells sorted from adipose tissue that came from six “aviremic” patients. Those results confirmed that adipose tissue constituted a viral reservoir, even in ART-suppressed patients [107].

Later, it was described that some viral proteins of HIV—such as Viral protein R (Vpr), the Trans-activator of the Transcription protein (Tat), and the Negative regulatory Factor (Nef)—could mediate changes in adipocyte function through the alteration of the expression of adiponectin, LPL, GLUT4, and PPAR-γ [112,113] (Figure 2B). This correlates with the lipodystrophy syndrome observed in some patients.

In animal models, with the transgenic overexpression of viral proteins, Vpr was shown to block preadipocyte differentiation and the PPAR-γ-expression of the glucocorticoid receptor in adipocytes and was found to inhibit PPAR-γ in hepatocytes [114].

Additionally, a different study conducted using human samples from HIV-infected and HIV-negative women reported that HIV altered fat architecture. This study found that HIV patients presented a higher number of smaller adipocytes and increased fibrosis characterized by compact and thick fibrils of collagen invading the intercellular area. Further investigation showed that an increase in the mRNA and protein levels of collagen 1 α2 induces collagen expression in adipocytes, which causes fibrosis. Moreover, the presence of lentiviral proteins such as Nef was associated with a decrease in cellular lipid accumulation and in the expression and protein level of the adipogenic markers PPAR-γ and fatty acid-binding protein 4 (FABP4) [115]. Altogether, this suggests that HIV is a pathogen capable of using adipose tissue as a reservoir, thus infecting T CD4 cells present in the adipose tissue, which, in the end, alters the metabolism of these patients.

### 3.3. Mycobacterium tuberculosis

In the case of bacteria, some reports also associate the presence of bacteria in adipose tissue with dysregulation and obesity.

*Mycobacterium tuberculosis* (Mtb) causes tuberculosis, which is one of the top infectious diseases in humans worldwide. According to the WHO, a quarter of the world’s population has been infected with this pathogen. The primary tissue infected by Mtb is the lung; however, it has been described that during a latent infection, when the symptoms of the disease are absent but the bacteria are present, Mtb can stay in extrapulmonary cells that function as reservoirs.

Among the Mtb reservoirs are adipocytes [116]. This tropism has been thoroughly described in mouse models. In addition, several studies found the presence of Mtb in the adipose tissue after infection with the use of aerosol doses or by using different infection routes such as intravenous or intranasal routes. In this case, Mtb was identified in visceral adipose tissue, in both the adipose fraction and in the SVF (macrophages) between 14 and 28 days post-infection, indicating that adipose cells and leukocytes present in fat tissue harbor Mtb [117,118]. In addition, Mtb can infect adipocytes by binding through scavenger receptors in both mice and humans [119].

Mycobacteria remain in a dormant state in adipocytes [120] and need fatty acids from the host to survive [120]. Once in the cells, Mtb dysregulates all the cell metabolisms. For example, it has been well-described how Mtb modifies lipid and glucose metabolism at the cellular level by differentiating macrophages into foamy cells [121].

In order to obtain free fatty acids during active growth in vitro and in vivo, Mtb upregulates neutral lipid stores, some genes related to the accumulation of lipid bodies, and genes related to triacylglycerol synthases and lipases [120] (Figure 2C). Additional evidence identified that Mtb infection in 3T3-L1 induced the downregulation of genes involved in de novo fatty acid synthesis and the upregulation of genes related to triglyceride biosynthesis [122]. Aside from affecting lipid metabolism, mice afflicted with low doses of aerosol infection also presented an inflammation process in visceral and brown adipose tissues due to higher macrophage recruitment [123].

Interestingly, the link between Mtb infection and the immune response in adipose tissue has been evaluated and was found to affect the production of pro-inflammatory cytokines (TNF-α, IL-6, MCP-1) and leptin, which induces metabolic dysfunction in obesity [117,118]. It was also observed that these infected adipocytes secreted more CCL5; this increase is associated with hypertrophy. Furthermore, after eight weeks of infection, the mice showed an increase in hypertrophy as well as an increase in hyperplasic cells [123]. The alteration of cytokines was also observed in rabbits infected with Mtb, and an increase in the levels of proinflammatory mediators such as leptin, TNF-α, IL-6, and PPAR-γ was found. All of this could lead to dyslipidemia [116].

Although the majority of the functional tests have been conducted in mice, there has been evidence of the presence and influence of Mtb in humans. For example, in 2006, Mtb was found in the biopsy of the adipose tissue of an infected patient. Additionally, the researchers found mycobacteria in the adipose tissue of patients who died of other causes, suggesting that the infection was persistent. Later on, the Mtb’s presence in adipocytes was shown in vitro and in adipose tissue biopsies from TB patients [123].

Overall, it is important to consider Mtb infection as a subjacent cause of obesity, as it could remain dormant for years in adipose tissue, altering cell metabolism, triggering potential low-grade inflammation, and later, an increase in obesity.

### 3.4. Chlamydia ssp.

The principal pathogen associated with urogenital infections and sexual transmission is the intra-cellular obligate bacterium *Chlamydia* ssp. [124]. Some evidence to establish the seroprevalences in obese humans showed a critical response to IgG and IgA for *Chlamydia pneumoniae*. As for the other pathogens that were previously described, an increase in circulating lipid levels has been observed in samples from Chlamydia-seropositive individuals, suggesting a connection with the possible development of obesity. The vital role of lipids as intracellular nutrients for the proliferation of Chlamydia has been documented [125,126]. A current in vitro study has shown the ability of Chlamydia to infect preadipocytes and promote their differentiation to adipocytes in the 3T3-L1 cell line, suggesting the possibility that Chlamydia can induce infectobesity [127]. The animal models support the idea that Chlamydia activates lipid pathways during infection. Mouse adipocytes infected with Chlamydia develop a stress response of the endoplasmic reticulum that culminates in an overproduction of reactive oxygen species and the accumulation of cytoplasmic calcium. These signals have been associated with the production and secretion of molecules such as FABP4 [128,129] (Figure 2D). FABP4 is a cellular factor that regulates fatty acid mobilization and activates pathways associated with inflammation and metabolisms. Several reports suggest the use of this protein as a marker of obesity [130]. The link between infection with Chlamydia and this pathway could be a rearrangement of the cellular distribution of lipids to help the intracellular proliferation of bacteria; however, more studies are necessary to support this idea.

### 3.5. Trypanosoma cruzi

It has also been described how some parasites are capable of infecting adipocytes, which is the case of *T. cruzi*. This pathogen is the etiological agent of Chagas disease, a zoonotic pathology that can be transmitted to humans by blood-sucking triatomine bugs and is mainly found in rural regions in America [131,132]. One of the most affected tissues in Chagas disease is the heart, which triggers inflammatory cardiomyopathy [133].

The lifecycle of *T. cruzi* occurs between an infected triatomine insects (vector) and a human (host) and consists of three different stages [131,132]. This parasite infects several cell types such as macrophages, fibroblasts, nerve cells, and muscle cells [132]. However, in recent years, the ability of *T. cruzi* to infect adipose tissue has been described together with its relationship with metabolic alterations [15]. During chronic infection, adipocytes may be considered crucial long-term reservoirs for *T. cruzi* [15].

Experimental reports have described how hyperglycemia could be related to higher parasitaemia and mortality in *T. cruzi*-infected mice [134]. In the case of in vitro experiments, it has been demonstrated that 3T3-L1 cells can be efficiently infected with *T. cruzi*, which modifies their metabolic and proteomic response. For example, *T. cruzi*-infected adipocytes switch the expression levels of adipocyte-specific or -enriched proteins, with adipokines being one of the main altered proteins. Then, *T. cruzi* infection guides adipokines to a unique metabolic signature [15]. Additionally, electron micrographs of *T. cruzi*-infected cells showed a significant part with intracellular parasites around the lipid droplets, which are considered master regulators of lipid and energy homeostasis [135].

During acute infection, glucose levels in all the *T. cruzi*-infected mice were below those measured in the controls. However, an oral glucose tolerance test indicated a relatively normal ability to clear ingested glucose despite the high degree of inflammation associated with this infection [134]. Moreover, it was observed during murine acute infection that the levels of adiponectin were diminished in mice infected with *T. cruzi*. Furthermore, the expression of PPAR-y was also low. In the same mice model, it was found that the changes in lipid regulators were associated with calcium control, as the Ca2þ/calcineurin/NFAT cascade was upregulated in adipocytes, thus correlating with the decrease of adiponectin in adipocytes infected with *T. cruzi* [136].

In mice, not only adipokines were dysregulated in the presence of *T. cruzi*, but so were signaling molecules stimulating inflammation such as Toll-like receptors (TLR-2 and -9). Additionally, cultured adipocytes presented a higher activation of the MAPK pathway [86]. Cytokines (TNF-α, IL-1β, and IFN-γ) and chemokines (CCL2, CCL4, CCL5, CCL6, CCL7, CCL8, and CCL12), as well as the levels of CXCL9 and CXCL10, were markedly elevated in the adipose tissue of acutely infected mice. The levels of IL-1a, IL-1b, IL-17b, and IL-18 were also increased [134,137] (Figure 2E). The infection also affected the amount of lipids in adipose tissue, with a 15% reduction in lipid content and a 25% loss of lipids in the adipose tissue being observed [137].

Continuing with the mouse model, in the case of chronic infection, it has been reported that the loss of fat cells in *T. cruzi*-infected mice produces an increase in cardiac lipid load and a deregulated cardiac lipid metabolism, which leads to mitochondrial oxidative stress and endoplasmic reticulum stress. In addition, the loss of fat cells has been found to increase cardiac parasite load during acute infection and to alter immune signaling in the hearts of infected mice during chronic infection [138]. Altogether, these results suggest that adipose tissue (fat tissue) and the liver play important roles in maintaining and regulating cardiac lipid metabolism.

In humans, a higher incidence of diabetes in Chagas-infected people has also been described. This relationship may be due to the reduction of insulin in infected people. Additionally, it is worth mentioning that the infection of adipocytes produces an increase in proinflammatory cytokines and chemokines and a decrease in adiponectin and PPAR-γ (negative regulators of inflammation). Diverse reports indicate that the lack of adiponectin produces a cardiomyopathic phenotype. It is possible that *T. cruzi* infection affects the heart via the lack of adiponectin during Chagas pathogenicity [15].

It is well-known that *T. cruzi* persists in adipose tissue, and that it is capable of altering the homeostasis of the adipose tissue by regulating lipolysis and inflammation, thus promoting weight gain and supporting the infectobesity theory.

### 3.6. Toxoplasma gondii

*Toxoplasma gondii (T. gondii)*, which is an intracellular protozoan that infects humans and domesticated and wild warm-blooded animals, is widely distributed around the world [139]. *T. gondii* infection is acquired by the ingestion of oocytes in contaminated water or vegetables, or by “fed” cysts in infected meat [139]. This parasite could pass through the intestinal epithelial barrier and multiply in the parasitophorous vacuole inside of various cell types [140]. The cells that can be infected by this parasite include dendritic cells, natural killers, monocytes, and macrophages [141,142]. This parasite can even cross the placental barrier and infect the fetus [143]. At the same time, the slow form of *T. gondii* (bradyzoites) can invade several tissues such as the heart, skeletal muscle, lung, brain [144,145], or adipose tissue. *T. gondii* has recently been described as a potential target in obesity research due to its implication in inflammation and its participation in the damage of pancreatic beta cells [146]. Those phenomena are related to metabolic syndrome. Additionally, the prevalence of metabolic syndrome was higher in patients who belonged to the *T gondii-*seropositive group than a *T. gondii*-seronegative group [147].

Experimental data show that *T. gondii* infection is related to a variation in weight gain; rats that were infected with *T. gondii* presented weight gain after 30 days. This was followed by a decrease in their weight in the following months. These weight variations were associated with behavioral changes due to the presence of *T. gondii* cysts in the brain, which possibly altered appetite regulation and hypothalamic function [148,149]. A study conducted on male Swiss Webster mice infected with *T. gondii* indicated that all infected mice presented a variation in weight, with an initial loss of 20–30%, and where half of these animals regained the lost weight or even more; however, another portion of these animals retained their low body weight [150]. Additionally, some authors mentioned that all these effects on weight variation may be influenced by the strain For example, it has been reported that two different strains of *T. gondii* had opposite effects on body weight [149,151].

In the case of human reports, there are some controversial studies. Thiodleifsson et al. reported a negative correlation between *T. gondii* and obesity; in this report, the authors did not find any correlation between the antibodies against *T. gondii* in overweight and non-overweight people [152]. However, Reeves et al. found a positive correlation between being overweight and *T. gondii* seroprevalence in adults (18–80 years) [149]. The variation in the results between these two investigations could be explained by the age range among the patients used by each independent study. Whereas the range in Reeves’ study was wider, Thiodleifsson excluded people older than 44 years old in their study.

These kinds of studies help to deepen our understanding of the relationship between obesity and infection. In addition, in the case of *T. gondii*, the presence of cytokines or immune molecules has recently been described. The presence of *T. gondii* leading to the release of IL-2 and other pro-inflammatory cytokines [153] could explain the weight variations present in *T. gondii*-infected people, i.e., they could be due to the presence of proinflammatory cytokines that induce adipocyte dysregulation [154]. The dysregulation of adipocytes then leads to metabolic syndrome due to an increase in the number of adipocytes and in insulin receptor sensitivity [154,155,156]. Additionally, adipocytes produce reactive oxygen species in response to inflammation and infections, including *T. gondii* infection [157,158,159,160] (Figure 2F). In general, the crosstalk between host immunity and parasites determines the dissemination of parasites and disease severity. However, further investigations are needed in order to fully understand the relationship between *T. gondii* and adipocytes and their association with overweight in rats and humans.

Altogether, the relationship between the previously described pathogens (viruses, bacteria, and parasites) and overweight demonstrates that adipocytes should be understood as cells associated with immunity. All of this is not strictly restricted to the pathogens mentioned here; we only focused on those for which more studies are to be conducted in humans. However, there are more pathogens associated with this theory in different animal models, and new ones that are capable of infecting human adipocytes are arising. Lastly, as this is a very complex field, and a detailed study of the pathogens needs to be conducted.

## 4. Gut Microbiota and Its Relationship with Obesity

In recent years, a natural component of our organism (the microbiota) has been considered as a principal regulator in different physiological processes. The microbiota is a community of microorganisms within a particular niche, whereas the consortium of all different microorganisms and their genomes is known as the microbiome [161,162]. Although bacteria have been identified within the main microorganisms that constitute this ecosystem, the participation of other biological entities such as archaea, eukaryotes (fungi), and viruses (bacteriophages) has begun to be considered [163,164,165]. Recent studies have revealed that these non-bacterial microbial populations are also dynamic communities, interacting with one another and playing a vital role in host wellbeing [165]. The gut microbiota has a wide distribution in our body, with a weight of 1–2 kg. In addition, the genetic information in this biological community is 10 times greater than the human genome [166]. Among the main functions of the gut microbiota is protecting and maintaining the intestinal mucosa through a symbiotic relationship with the host; additionally, it plays important roles such as supporting the immune development and providing protective, structural, and metabolic functions essential for the human body [165]. This intimate relationship allows microorganisms to contribute to physiological homeostasis, and abnormalities in the microbiota participate in the development of various diseases [167]. In this sense, it has been proven that there is an essential connection between metabolic disorders (including obesity) and the gut microbiota by showing that there is a regulation of fat storage, thus increasing energy collection and regulating the formation of substrates for lipid storage [166,168,169].

Studies to understand the relationship between the microbiota and obesity have identified a different microbial diversity in the obese population, with a higher proportion of Firmicutes bacteria and a low amount of Bacteroidetes compared to the microbiota of people with healthy weights [168] (Figure 3A). The main microbes identified in the microbiota of the obese population include *Clostridium innocuum*, *Eubacterium dolichum*, *Catenibacterium mitsuokai*, *Lactobacillus reuteri*, *Lactobacillus sakei*, and Actinobacteria, while Archaea such as *Methanobrevibacter smithii* have also been identified, albeit in lower numbers [169,170]. Therefore, it has been strongly suggested that a high proportion of the metabolic products and other biomolecules produced by these bacterial consortia could be responsible for the molecular mechanisms that link obesity and the microbiota [171].

It has been documented that the abundance of Bacteroidetes in the intestinal microbiota has a direct positive relationship with the concentrations of short-chain fatty acids (propionate, butyrate, and acetate) in feces; these are associated with fermentation processes or the absorption of carbohydrates from the diet [172]. Moreover, these metabolites play a role in cellular signaling mediated by interactions with G-protein-coupled receptors, which have been reported to be widely distributed in adipocytes [173]. Interestingly, the receptor GPR43, when interacting with a short-chain fatty acid, participates in the modulation of the immune response by reducing the release of inflammatory mediators and increasing hypothalamic sensitivity to leptin. In addition, in adipose tissue, the activation of GPR43 could suppress insulin signaling, culminating in the inhibition of fat accumulation and activating lipolysis in adipocytes (Figure 3B) [174]. Finally, animal models deficient in the expression of GPR43 that were subjected to a diet high in carbohydrates and fats developed less body mass and higher lean mass than wild-type animals [175].

Another microbiota product that may be a factor related to obesity is lipopolysaccharide (LPS), an amphipathic molecule mainly consisting of an O-antigen, a core region, and a lipophilic lipid (lipid A) that has toxic effects and immunomodulatory functions [176]. Studies with animals subjected to high-fat diets have identified an increase in the presence of Gram-positive bacteria such as Bifidobacterium in the intestinal gut and high levels of LPS in the plasma of these animals. Additionally, a mechanism for the presence of LPS in the circulation has been evidenced to be associated with a decrease in Bifidobacterium, which is related to the low production of a proglucagon-derived peptide in the intestine with trophic action (GLP-2) [177]. This molecule is produced by the L cells in the intestinal mucosa, participates in intestinal growth, and functions as a barrier. Its decrease promotes the lowering of the expression of tight-junction proteins such as zona occludens proteins, which promotes an increase in intestinal permeability and thus facilitates the translocation of LPS to the circulation [177]. However, for the LPS to regulate adipose tissues, it needs to be transported in the serum to these areas. It has been identified that there is a primary interaction of 90% of LPS with lipoproteins [178]. One of the proteins with the most significant capacity to bind to LPS is high-density lipoprotein (HDL). In addition, it has been observed that LPS also binds rapidly to soluble CD14 or to LPS-binding proteins (LBPs), which are located within HDL and thereby transport the LPS to fat tissue (Figure 3C). Finally, this complex, HDL–sCD14–LB–LPS, may be internalized by the adipocytes and macrophages [179,180].

To better understand the possible role of LPS as a stimulus in adipocytes, several hypotheses are being investigated. One of them is related to a possible effect of pyroptosis generation and inflammation mediated by inflammasome activation [181]. In animal models, it has been identified that intracellular LPS can activate caspase-11, whose homolog in humans is caspase-4/5 [182]. Furthermore, it has been described that this non-canonical pathway can serve as a sensor of the innate response independent of the effect of LPS recognition by TLR-4 [183]. Evidence has suggested that there is an LPS-binding site within the CARD domain of caspase-11 (caspase-4/5), and it has also been identified that LPS binding promotes caspase oligomerization, promoting NLRP-3-mediated assembly by ASC. Therefore, it is possible that NLRP-3 activates caspase-1, which leads to the activation of pro-IL-1b and pro-lL-18, and finally, to the secretion of these proinflammatory cytokines. However, LPS-activated caspase-4/5/11 can also directly induce pyroptosis [184].

Interestingly, it has been identified that in the adipose tissue of obese people, there are cellular structures made up of adipocytes, macrophages, and the remnants of adipocytes that have undergone pyroptosis. These crown-like structures can hold macrophages in adipose tissue, thereby initiating insulin resistance [185]. The hypothesis on the role of LPS in the generation of pyroptosis in adipocytes is associated with the possible aggregation of LPS in the adipocyte membrane and its interaction with pro-caspases-4, 5 and 11; through the CARD domains, the LPS can associate and mediate the oligomerization and activation of caspases, culminating in the activation of pyroptotic pathways [181].

The emerging relationship between obesity and the microbiota opens a new and interesting landscape for investigation, given that the microbiota is not widely considered from a pathogenic perspective; a slight imbalance could lead to obesity. For this reason, further research could help to elucidate the association between microbiota stimulation, immunity, and infectobesity.

## 5. Conclusions

The immediate solution for managing obesity in the population is the promotion of weight control and weight loss. However, the multifactorial etiology that triggers this condition in human health is complex. It involves new points that must be addressed, such as the possible interaction between some infections and the development of obesity. In this respect, the evidence shows that adipocytes in some infections have been correlated with a trigger for obesity. These cells could function as a potential target for infection, allowing pathogens to access this tissue and generate different events that lead to the activation of lipogenic pathways, adipocyte differentiation, and the proliferation process. Additionally, the possibility that adipocytes could function as reservoirs for these pathogens could be another mechanism by which infections in the obese population exhibit persistent behavior and produce more severe outcomes.

Interestingly, a link between obesity and infections is the intermediary role of the immune response of the host. It is clear that the notion that adipocytes function simply as energy reservoirs is incorrect. These cells have great plasticity, allowing them to sense inflammatory signals and produce molecules that function as immune stimulators. The infectobesity hypothesis should consider future interrogatives to help elucidate how the immune response regulates inflammatory processes within the adipocyte under infection conditions. These signals could produce a favorable cellular or organism environment, which may hamper the control of the infection or trigger mechanisms that generate a type of persistence in these tissues, thus favoring an increase in fat cells and initiating the obesity process. It is necessary to extend and address this interaction.

## Figures and Tables

**Figure 1 ijms-23-06154-f001:**
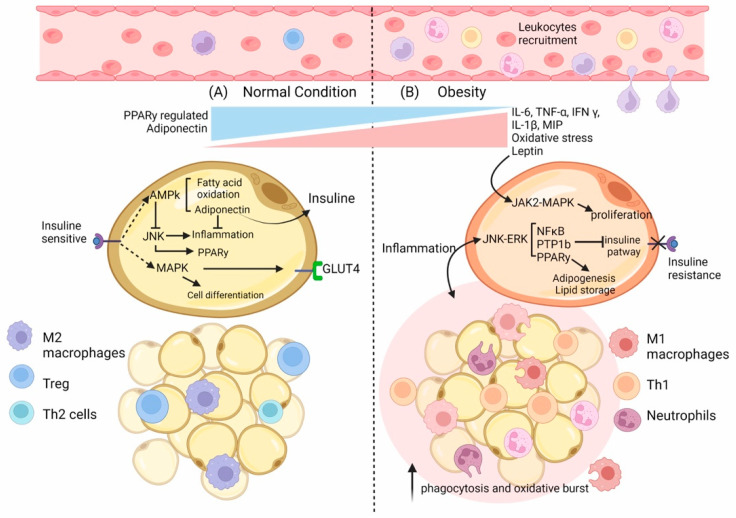
**Environment state and adipocyte’s changes in normal vs. obese conditions.** (**A**) In normal conditions in the adipose tissue (AT), M2 macrophages are distributed evenly throughout the tissue. Lean AT also contains CD4+ T cells—TH2 and Treg cells—and preadipocytes that contribute to the anti-inflammatory and insulin-sensitive state. As a result, adipokines such as adiponectin are increased, whereas leptin content is decreased. In adipocytes, the regulation of differentiation and proliferation occurs by activating the insulin receptor (IR), which also controls the incorporation of glucose. Moreover, IR regulates the activation of AMPK in an energy state-dependent manner, promoting β-oxidation and adiponectin expression, resulting in the control of inflammation. (**B**) In obese AT, specific immune cells such as macrophages increase their number by local proliferation and contribute to secreting pro-inflammatory cytokines and ROS. In addition, the content of other immune cells also changes with an increase in Neutrophils, CD4+ Th1, and CD8+ T cell numbers, a reduction in Treg cell as well as an increase in the number of preadipocytes. As a result, the production of pro-inflammatory adipokines such as TNF-α, IL-6, IL-1β, and MCP1 increases; consequently, there is an increased immune cell infiltration and a polarization of the phenotype. Furthermore, in adipocytes, the augmented inflammatory state disrupts the regulation of signaling pathways such as JNK-ERK, which involves the inhibition of the IR by PTPB1, the up-regulation of PPAR-γ, and the synthesis of lipids as well as of the inflammatory NFkB pathway, which is also enhanced by overexpressed Leptin in an obese condition.

**Figure 2 ijms-23-06154-f002:**
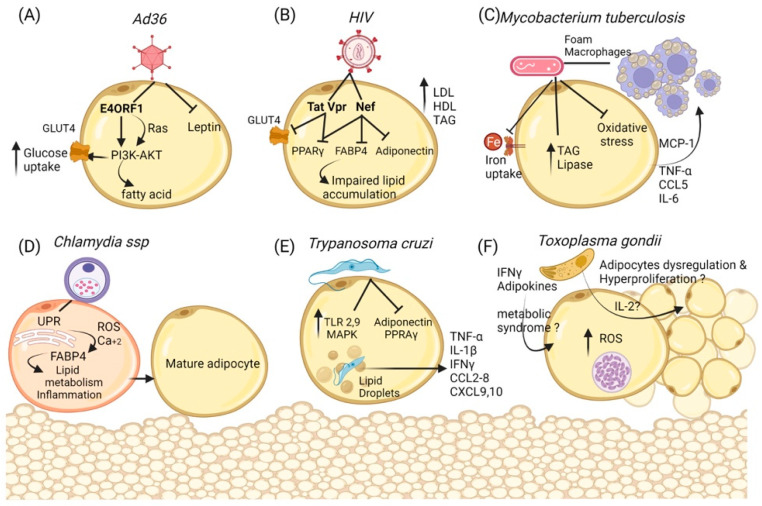
**Cellular and metabolic changes in adipocytes during infections.** (**A**) **Ad36** regulates the activation of Phosphoinositide 3-kinase (PI3K) by the recruitment of P13K to the membrane, PI3K-AKT leads to cellular glucose uptake by the glucose transporter Glut4 and could induce the de novo synthesis of fatty acids despite the down-regulation of the Insulin Receptor. Moreover, the inhibition of Leptin may increase lipid accumulation and the prevalence of obesity. (**B**) **HIV** proteins Vpr, Tat, and Nef participate as repressors of PPAR-γ and its target, such as GLUT4, besides Vpr expression, thus accelerating lipolysis and insulin resistance as well as preventing the maturation of adipocytes. (**C**) ***Mycobacterium tuberculosis*** can use adipose tissue as a reservoir and as macrophages, inducing lipogenic metabolism that can accumulate lipids (foam macrophages). Furthermore, adipocytes regulate inflammation by reducing oxidative stress in a decreased iron uptake manner, but augmenting pro-inflammatory cytokines and chemokines that recruit cells similar to macrophages. Additionally, the synthesis of triacylglycerol and lipase is increased to dispose of free fatty acids. (**D**) ***Chlamydia ssp*** infection increases the unfolded protein response and the guides to ROS production and Ca++ release; this could lead to FABP4 activity in adipocytes. Moreover, the expression of adipogenic markers is accompanied by lipid droplet accumulation, suggesting their transformation into differentiated adipocytes. (**E**) ***Trypanosoma cruzi*** infection dysregulates some proteins in the adipocyte, reducing adiponectin and PPAR- γ, but also signal molecules that stimulate inflammation such as Toll-like receptors (TLR-2 and -9). Additionally, cultured adipocytes presented a higher activation of the MAPK pathway in cultured adipocytes. Cytokines (TNF-α, IL-1β, IFN-γ) and chemokines (CCL2-CCL8 and CCL12), as well as the levels of CXCL9 and CXCL10, were markedly elevated in the adipose tissue of acutely infected mice. (**F**) ***Toxoplasma gondii*** bradyzoites use the adipose tissue as reservoirs; the infection with *T gondii* can induce oxidative stress and the subsequent inflammatory response. In addition, several cytokines, such as IL-2, could affect the adipocyte tissue and alter their metabolism or promote cell proliferation and hypertrophy as well the presence of Interferon-gamma and adiponectin chemerin in the serum, thus encouraging the development of the metabolic syndrome.

**Figure 3 ijms-23-06154-f003:**
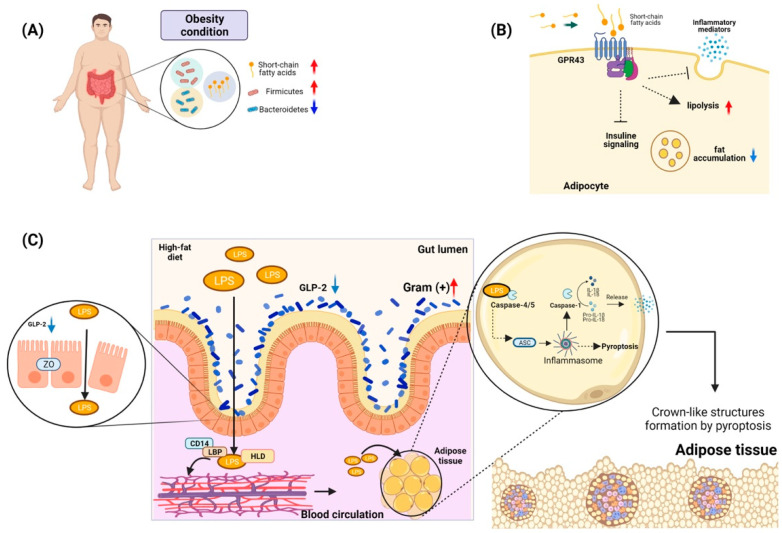
**Gut microbiota and its role in obesity**. (**A**) Phylum of bacteria related to the development of obesity in individuals. Under conditions of obesity, bacteria belonging to the phylum Firmicutes are increased, while Bacteroidetes are decreased. These bacteria have been related to the presence of short-chain fatty acids. (**B**) Some G protein-coupled receptors abundant in adipocytes, such as GPR43, may play a role in the modulation of the immune response. Activation of GPR43 can promote lipolysis and decrease fatty acid accumulation. (**C**) Antigens from the bacterial microbiota of the gut in conditions of obesity, such as LPS, can cross the epithelial barrier of the gut due to the rupture of tight junctions such as zona occludens (ZO), which is related to the decrease in the expression of GLP-2. This LPS can travel through the circulation by forming a complex of HLD plus LBP and CD14. This complex gives it hydrosoluble characteristics, allowing it to reach the adipose tissue to be absorbed by adipocytes. Once inside the adipocyte, it is recognized by Caspases 4/5, which trigger the activation of the inflammasome. Then, it could activate either the maturation of proinflammatory molecules such as IL-1B and IL-18 or activate pyroptosis. The pyroptosis present in the adipose tissue can trigger the formation of crown-like structures, which are composed of immune system cells such as macrophages and adipose tissue cells and cells in pyroptosis. These structures have been described as an essential element in insulin resistance and in the development of chronic inflammation in obese individuals.

## Data Availability

Not applicable.

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
