# Peer review of "The Immune Response in Adipocytes and Their Susceptibility to Infection: A Possible Relationship with Infectobesity"

_ijms, 2022, doi:10.3390/ijms23116154_

Round 1

Reviewer 1 Report

TO THE AUTHORS

Orestes López-Ortega 1, Nidia Carolina Moreno-Corona 2, Victor Holguin-Cruz 3,4, Luis Didier Garcia-Gonzalez 3, Addy Cecilia Helguera-Repetto 3, Mirza Romero-Valdovinos 5, Haruki Arevalo-Romero 6, Leticia Cedillo-Barron 4 and Moisés León-Juárez

The authors in this review aim to underline the interesting relationship between obesity in and infections. They hypothesized that a relationship could exist between infection adipose cells and metabolism. This is of great interest as it is still unclear the role of adipose tissue in the onset and infection progression. Moreover, as reader, I could not appreciate it. Despite it is interesting topic also for clinical changing I think that in its present form this review has many limitations: it appears not clear to the reader, since the beginning in the abstract. Also some limitations are present related to the structure and to the consistence of the article, to the order of the sections, to the spelling mistakes and to the references. Lastly, it is not clear why the authors have chosen some microorganism instead others. Maybe this topic is so complex that could be not treated in an exhaustive way in one review.

Abstract:

It is not clear and does not respect the structure of the whole review. For example, this sentence is not clear: “In addition, there is a lot of evidence that indicates that there is an exacerbated functioning of the immune response in obesity, which has a negative impact on infections in adipose tissue

Introduction; It is confused and not clear. It appears as several sentences written without a coherent general and conclusive message to the reader.  Furthermore, references are twice repeat without sense.

Review Sections: in several part of the different described sections some concepts are repeated. Also the order of the sections could be reviewed. Firstly, the anatomic composition of the adipose tissue, the adipocytokines produced and afterwards the adipocytes function and their relationship with the immune cells and with the potential microorganism inducing infections, should be described. Afterwards, the contribute of microorganism as virus, bacteria, protozoa presented are not conclusive in a real take home message to the reader.

Infectobesity: in this paragraph, interesting and new, references are totally lacking.

References: many references are old and not pertinent to the text. For examples, reference number 1 is not related to the author’s sentences and it is too old (21 years ago) in comparison to the large amount of literature recent about the very interesting topic treated.

Spelling mistakes: often the sentences are not correct or not clear, also for acronym, for example, what is the acronym VIH related to leptin infections?

Author Response

Reviewer 1:

The authors in this review aim to underline the interesting relationship between obesity in and infections. They hypothesized that a relationship could exist between infection adipose cells and metabolism. This is of great interest as it is still unclear the role of adipose tissue in the onset and infection progression. Moreover, as reader, I could not appreciate it. Despite it is interesting topic also for clinical changing I think that in its present form this review has many limitations: it appears not clear to the reader, since the beginning in the abstract. Also some limitations are present related to the structure and to the consistence of the article, to the order of the sections, to the spelling mistakes and to the references. Lastly, it is not clear why the authors have chosen some microorganism instead others. Maybe this topic is so complex that could be not treated in an exhaustive way in one review.

1)Abstract:

It is not clear and does not respect the structure of the whole review. For example, this sentence is not clear: “In addition, there is a lot of evidence that indicates that there is an exacerbated functioning of the immune response in obesity, which has a negative impact on infections in adipose tissue”

We are very grateful with this comment, we made an extensive revision of the text, and we corrected all the sentences that we consider confusing. We highlighted in the text by red color.

  • We changed the phrase “This suggests that the factors involved in this condition need to be reconsidered” for: This suggests that the factors involved in this condition need to be reconsidered, as, some new factors are arising as etiological cause of this disease.
  • We changed the phrasethe progress of obesity with some infections, and adipose tissues could either be reservoirs for these pathogens or play an active role in this process.” For: the progress of obesity with some infections, and the functions of adipose tissues, which involve cell metabolism and adipokine release, among others; furthermore, recently, it has been described that adipocytes could either be reservoirs for these pathogens or play an active role in this process
  • We changed the phrase In addition, there is a lot of evidence that indicates that there is an exacerbated functioning of the immune response in obesity, which has a negative impact on infections in adipose tissue” was changed for: In addition, there is diverse evidence indicating that, during obesity, the immune system is exacerbated, suggesting an increased susceptibility of the patient to the development of several forms of illness or death. And we put it in a different place in the abstract.

2)Introduction:

It is confused and not clear. It appears as several sentences written without a coherent general and conclusive message to the reader.  Furthermore, references are twice repeat without sense.

A: Thank you very much for this comment, we also review the structure of the introduction, and we change the following:

  • -“Obesity is becoming a growing health problem worldwide; we are facing an obesity epidemic with a complex and multifactorial etiology” for:
    • We are facing an obesity epidemic, as obesity is becoming a growing health problem worldwide, with a complex and multifactorial etiology.
  • -“ Several databases focused on the obesity research field show that susceptibility to infections and obesity are related. These databases also point to adipocytes as key components in the crosstalk between adipose tissue, immune cells and microorganisms.” For:
    • Data from several databases focused on the obesity research field, indicate that susceptibility to infections and obesity are related. These databases also place adipocytes as key components in the crosstalk between microorganism, obesity, and immune cells
  • -“This is considered the "infectobesity hypothesis". This theory is sustained by several facts: first, it has been described how the infection of pathogens alone impacts the architecture of adipose tissue due to inflammation or pathogen persistence”; For:
    • opening a new landscape in this research field and leading to the emergence of a new "infectobesity theory". This theory is sustained by multiple lines of evidence: Firstly, many reports have described how infections by pathogens alter the architecture of adipose tissue due to inflammation or pathogen persistence
  • We added; e signalling pathways after the phrase “since some pathogens are able to dysregulate components
  • We also include this phrase “with adenovirus 36 being one of the most studied pathogens associated with obesity.” after This theory maintains that some microorganisms are capable of inducing the obesogenic process.
  • The phrase “Excessive adipose tissue is associated with several chronic diseases” was deleted from the text”
  • “neuronal nature” was changed for “neuronal origin”
  • “adipocytes and can be divided into white and brown adipose tissue” was changed for: adipocytes and can be divided into white, beige and brown adipose tissue
  • The phrase: but excessive adipose tissue is associated with several chronic diseases was added at the end of the paragraph: body temperature by thermogenesis

  • The phrase: Moreover, in obesity, it has been shown that the state of low-grade inflammation, was changed for: “Moreover, obesity has been considered as a state of low-grade inflammation”

3)Review Sections:

In several part of the different described sections some concepts are repeated. Also the order of the sections could be reviewed. Firstly, the anatomic composition of the adipose tissue, the adipocytokines produced and afterwards the adipocytes function and their relationship with the immune cells and with the potential microorganism inducing infections, should be described. Afterwards, the contribute of microorganism as virus, bacteria, protozoa presented are not conclusive in a real take home message to the reader.

A: We appreciate these comments, we reorganize the sections according to the suggestions of the reviewer, then we added a conclusion in each section (microorganism and humoral sections) and also a general conclusion at the end of this review.

  • We changed the order and add a special section for:  Adipose tissue´s composition
  • We added this phrase at the beginning: Adipose tissue is a dynamic organ distributed throughout the body.
  • We include this new paragraph for more detail about AT composition:
    • Adipose tissue can be classified by morphology into white, brown or beige subsets, and is organized into discrete depots throughout the body [17,18]. Brown and beige adipocytes represent a small proportion of the total adipose tissue [18], with white ad-ipose tissue (WAT) being the most abundant form, found in almost every area of the body. The major WAT depots can be broadly classified by location as either subcutaneous (under the skin) or visceral and visceral/omental (located intra-abdominally, ad-jacent to internal organs) [17,18].   Although adipocytes account for most of the volume of AT, they only make up about 50% of the cellular content [16]. Other cell types including endothelial cells, fibroblasts, and immune cells are present in this tissue [17]. Macrophages and T lymphocytes are the principal immune cells infiltrating adipose tissue, and one of their functions is to secrete proinflammatory mediators [19,20]. In more detail, visceral AT, in contrast to subcutaneous AT, tends to have a higher content of macrophages, regulatory T cells, natural-killer T cells, and eosinophils [16], suggesting it could be a potential immunologic effector site, but more studies need to be conducted to position adipose tissue as a new immune site.
  • We include this paragraph at the beginning of the next section adipokines: Adipokines are a family of hormones and cytokines that are secreted by adipose tissue and can modulate diverse cellular responses in immune cells, such as cell migration [21], or even the cellular subpopulations of T cells [22]
  • Then we move the section leptin and afterwards we put a subsection for adiponectin.
  • Also we add the next phrase: In fact, older ICAM-1-deficient mice or even those lacking CD11b/CD18 (one of the main ligands for ICAM-1) show an increase in obesity” after this paragraph Adiponectin presents specific anti-inflammatory properties through the inhibition of IL-6 production or by the induction of anti-inflammatory cytokines, such as the IL-10 or IL-1 receptor antagonist, and the reduction in ICAM-1 and VCAM-1
  • We call the next subsection: Adipocyte mass and function and their relationship with the immune cells
  • We reorganize the section starting with a brief introduction of immune function in adipose tissue and afterwards we put the paragraphs dedicated to the main signalling pathways used for the immune function in the adipocytes
  • We added the following conclusion to this subsection:
    • In general, diverse signaling pathways are activated by either microorganisms or immune cues affecting adipocytes’ homeostasis; this convergence highlights the infectobesity hypothesis as a new research field.
  • The next subsection: IFN Response we add an introductroy paraghraph:
    • As previously mentioned, one of the most altered signaling pathways is JAK–STAT, one of the most important downstream effectors being interferons (IFNs).
  • For the next section: Adipocytes as Infection Targets and Their Relationship with Infectobesity, we also reorganized the introductory part by adding the following phrase: As mentioned previously, adipose tissue plays a vital role in immunity and the inflammation mediated by infiltrated or resident immune cells or by humoral cues like cytokines or adipokines
  • We also mention general cases of adipose tissue during infection:
    • Diverse signaling pathways are affected in adipose tissue in response to infection, with the PI3K–AKT–mTOR axis being one of the most altered. This signaling pathway is involved in the control of nutrients, cellular proliferation, growth, survival and mobility [82,83]. In the case of infections, PI3K–AKT–mTOR is a target for the modulation of expression or state of activation. For example, T. cruzi infection increases PI3K expression and AKT activation in 3T3-L1 adipocytes [84], linking T. cruzi infection with the insulin–IGF-1 receptor cascade and, later, with insulin resistance.
    • Additionally, leptin is essential for negatively modulating the immune response against infection; for example, the absence of leptin in OB/OB mice leads to a higher formation of granulomas in mice infected with mycobacteria. Moreover, an increase in leptin receptors has been described in HIV infections [24]. Additionally, active tuberculosis is correlated with weight loss, cachexia, and low serum concentrations of leptin, which, in turn, suppresses lymphocyte stimulation and the secretion of Th1 cytokines such as IL-2, IFN-γ and TNF-α [24].
  • At the end of this subsection we added the following conclusive message: Therefore, the next section focuses on diverse pathogens that have demonstrated tropism for adipocytes, some of which are arising as new candidates for adipocyte dysregulation.
  • For each pathogen we include a more clearly take-home message.
  • For AD-36 we include the next phrase:
    • All this implies that Ad-36 infection leads to adipogenesis in mice; however, in humans, this hypothesis needs further research. Although the majority of the papers describe Ad-36 as the main actor in the weight gain by producing adipocyte hypertrophy, the complexity of this phenomenon based on the relationship of immune cells, adipocytes and Ad-36 is not fully understand; a network between these three components as a possible cause for obesity may exist.
  • For HIV we improve the conclusion that was mentioned in the text:
    • Altogether, this suggests that HIV is a pathogen capable of using adipose tissue as a reservoir and furthermore infecting T CD4 cells present in adipose tissue, which, in the end, alters the metabolism of these patients.

  • For Mycobacteria tuberculosis we include the next paragraph:
    • Overall, it is important to consider Mtb infection as a subjacent cause of obesity, as it could remain dormant for years in adipose tissue, altering cell metabolism, triggering potential low-grade inflammation and, later, an increase in obesity.

  • For Chlamydia sp. We highlighted this phrase: The link between the infection with Chlamydia and this pathway could be a rearrangement of the cellular distribution of lipids to help the intracellular proliferation of bacteria; however, more studies are necessary to support this idea., as we consider it as a take-home message.

  • For Trypanosoma cruzi we added the next phrase:
    • It is well known that T. cruzi persists in adipose tissue and that it is capable of altering the homeostasis of adipose tissue by regulating lipolysis and inflammation, promoting weight gain, supporting the infectobesity theory.

  • For Toxoplasma gondii We highlighted this phrase: In general, the crosstalk between host immunity and parasites determines the dissemination of parasites and disease severity. However, further investigations are needed in order to fully understand the relationship between T. gondii and adipocytes and their association with overweight in rats and humans, as we consider it as a take-home message.

  • As a final conclusion for this section we added the following:.
    • Altogether, the relationship between the pathogens described previously (viruses, bacteria and parasites) and overweight demonstrates that adipocytes should be understood as cells associated with immunity. All of this is not strictly restricted to the pathogens mentioned here; we only focused on those for which more studies are to be conducted in humans. However, there are more pathogens associated with this theory in different animal models, and new ones that are capable of infecting human adipocytes are arising. Lastly, as this is a very complex field, a detailed study of the pathogens needs to be conducted.

4) Infectobesity: in this paragraph, interesting and new, references are totally lacking. References: many references are old and not pertinent to the text. For examples, reference number 1 is not related to the author’s sentences, and it is too old (21 years ago) in comparison to the large amount of literature recent about the very interesting topic treated.

We have understood the request of the reviewers and updated references have been added to the work.

5) Spelling mistakes: often the sentences are not correct or not clear, also for acronym, for example, what is the acronym VIH related to leptin infections?

We appreciate the reviewer's comments, and we have changed and revised these errors in addition to the proofreading and grammar service that have been of great help in polishing this manuscript.

Reviewer 2 Report

The study of the relationship between pathogens and obesity is an emerging field of medical research, and it has been identified with the term of infectobesity. 

The review proposed by Lopez-Ortega and colleagues accurately addresses the topic.

I would like to suggest to include a paragraph about gut microbiota and to discuss in the conclusion paragraph (or in a dedicated paragraph) the causal relationship between infections and obesity.

Author Response

Reviewer 2:

The study of the relationship between pathogens and obesity is an emerging field of medical research, and it has been identified with the term of infectobesity. 

The review proposed by Lopez-Ortega and colleagues accurately addresses the topic.

I would like to suggest including a paragraph about gut microbiota and to discuss in the conclusion paragraph (or in a dedicated paragraph) the causal relationship between infections and obesity.

We appreciate the reviewer's comment. We believe that this recommendation is correct, so we have integrated a section on gut microbiota into the manuscript, which has been related to obesity as a possible additional factor affecting adipocyte physiology. We added a figure to exemplify diverse microbiota factors driving to infectoobesity.

  • We include the following section for gut microbiota: In recent years, a natural component in our organism (known as the microbiota) has begun to be considered as a principal regulator in different physiological processes. The microbiota is a consortium of microorganisms engaged in a multifaceted relationship with the host to maintain balance for mutual benefit [159]. Although bacteria have been identified within the main microorganisms that constitute this community, the participation of other biological entities such as bacteriophages and fungi has begun to be considered [160,161]. An example of this community is the gut microbiota, which has a wide distribution in our body, with a weight of 1-2 kg. In addition, the genetic information in this biological community is 10 times greater than the human genome [162]. Among the main functions of the gut microbiota is protecting and maintaining the intestinal mucosa, through a symbiotic relationship with the host. This intimate relationship allows microorganisms to contribute to physiological homeostasis, and abnormalities in the microbiota participate in the development of various diseases [163]. In this sense, it has been proven that there is an essential connection between metabolic disorders, including obesity, and the gut microbiota, by showing that there is a regulation of fat storage, increasing energy collection, and regulating the formation of substrates for lipid storage [162,164,165].

Studies to understand the relationship between the microbiota and obesity have identified a different microbial diversity in the obese population, with a higher proportion of Firmicutes bacteria and a low amount of Bacteroidetes compared to the microbiota of people with healthy weights [166]. The main microbes identified in the microbiota of the obese population include Clostridium innocuum, Eubacterium dolichum, Catenibacterium mitsuokai, Lactobacillus reuteri, Lactobacillus sakei, and Actinobacteria, while Archaea such as Methanobrevibacter smithii have also been identified, albeit in lower numbers [167,168]. Therefore, it has been strongly suggested that a high proportion of the metabolic products and other biomolecules produced by these bacterial consortia could be responsible for the molecular mechanisms linking obesity and the microbiota [169].

It has been documented that the abundance of Bacteroidetes in the intestinal microbiota has a direct positive relationship with the concentrations of short-chain fatty acids (propionate, butyrate and acetate) in feces; these are associated with fermentation processes or the absorption of carbohydrates from the diet [170]. Moreover, these metabolites play a role in cellular signaling mediated by interactions with G-protein-coupled receptors, which have been reported to be widely distributed in adipocytes [171]. Interestingly, the receptor GPR43, when interacting with a short-chain fatty acid, participates in modulating the immune response by reducing the release of inflammatory mediators and increasing hypothalamic sensitivity to leptin. In addition, in adipose tissue, the activation of GPR43 could suppress insulin signaling, culminating in the inhibition of fat accumulation and activating lipolysis in adipocytes [172]. Finally, animal models deficient in the expression of GPR43 that were subjected to a diet high in carbohydrates and fats developed less body mass and higher lean mass than wild-type animals [173].

Another microbiota product that may be a factor related to obesity is lipopolysaccharide (LPS), an amphipathic molecule mainly consisting of an O-antigen, a core region and a lipophilic lipid (lipid A) that has toxic effects and immunomodulatory functions [174]. Studies with animals subjected to high-fat diets have identified an increase in the presence of Gram-positive bacteria such as Bifidobacterium in the intestinal gut and high levels of LPS in the plasma of these animals. Additionally, a mechanism for the presence of LPS in the circulation has been evidenced that is associated with a decrease in Bifidobacterium, which is related to low production of a proglucagon-derived peptide in the intestine with trophic action (GLP-2) [175]. This molecule is produced by the L cells in the intestinal mucosa, participates in intestinal growth and functions as a barrier. Its decrease promotes a low in the expression of tight-junction proteins such as zona occludens proteins, promoting an increase in intestinal permeability and thus facilitating the translocation of LPS to the circulation [175]. However, for the LPS to regulate adipose tissues, it needs to be transported in the serum to these areas. It has been identified that there is a primary interaction of 90% of LPS with lipoproteins [176]. One of the proteins with the most significant capacity to bind to LPS is high-density lipoprotein (HDL). In addition, it has been observed that LPS also binds rapidly to soluble CD14 or to LPS-binding proteins (LBPs), which are located within HDL and thereby transport the LPS to fat tissue. Finally, this complex, HDL–sCD14–LB–LPS, may be internalized by the adipocytes and macrophages [177,178].

To better understand the possible role of LPS as a stimulus in adipocytes, several hypotheses are being investigated. One of them is related to a possible effect of pyroptosis generation and inflammation mediated by inflammasome activation [179]. In animal models, it has been identified that intracellular LPS can activate caspase-11, whose homolog in humans is caspase-4/5 [180]. Furthermore, it has been described that this non-canonical pathway can serve as a sensor of the innate response independent of the effect of LPS recognition by TLR-4 [181]. Evidence has suggested that there is an LPS-binding site within the CARD domain of caspase-11 (caspase-4/5), and it has also been identified that LPS binding promotes caspase oligomerization, promoting NLRP-3-mediated assembly by ASC. Therefore, is it possible that NLRP-3 activates caspase-1, leading to the activation of pro-IL-1b and pro-lL-18 and, finally, the secretion of these proinflammatory cytokines. However, LPS-activated caspase-4/5/11 also can directly induce pyroptosis [182].

Interestingly, it has been identified that, in the adipose tissue of obese people, there are cellular structures made up of adipocytes, macrophages and the remnants of adipocytes that have undergone pyroptosis. These crown-like structures can hold macrophages in adipose tissue, thereby initiating insulin resistance [183]. The hypothesis of role of LPS in the generation of pyroptosis in adipocytes is associated with a possible aggregation of LPS in the adipocyte membrane and interaction with pro-caspases-4, 5 and 11; through the CARD domains, the LPS can associate and mediate the oligomerization and activation of caspases, culminating in the activation of pyroptotic pathways [179].

The emerging relationship between obesity and the microbiota opens a new and interesting landscape for investigation, considering that the microbiota is not widely considered from a pathogenic perspective; a slight imbalance could lead to obesity. For this reason, further research could help to elucidate the association of microbiota stimulation, immunity and infectobesity.

Round 2

Reviewer 1 Report

I thank the authors for reviewing the manuscript. Now it appears clearer, more complete and with the references updated. Furthermore, the gut microbiota section adds a very interesting scientific perspectives and strengthens the message of this reviewer.  At the same time it needs to be improved for some comments. Please, find the following comments.

Abstract: please, change word diverse with another one

Introduction: adipose tissue description: it is not made only by white, brown and beige, but also pink adipocytes, please add also reference about it (for example, see the paper, Corrêa LH, Heyn GS, Magalhaes KG. The Impact of the Adipose Organ Plasticity on Inflammation and Cancer Progression. Cells. 2019 Jun 30;8(7):662. doi: 10.3390/cells8070662).

Avoid to repeat references in this section (7,8 for example).

Review section

The Role of Adipose Tissue in the Immune Response,

Adipose tissue can be classified by morphology into white, brown or beige subsets

The authors already described it. Please, delete it previously or here.

Gut microbiota and its relationship with obesity,

At the beginning, please, describe better gut microbiota (it is not simply a natural component in our organism, as described), that is a complex microbial ecosystem that included other than bacteria, also virus, fungi and protozoa and parasites that play a fundamental role in immune system modulation and in maintaining good health for the host.

Please, correct the word adipocyte in the figure3 B.

Acronym, please, specify, when it is possible The cJun-N-terminal-kinase (JNK), PTP1b, PPAR-γ, Mtb, and all the others present in this section, the first time that you use these terms.

On the other hand IFNs is twice defined. Please, correct it.

Please, replace term cues, it appears not clear.

These sentence is not clear displayed in this way: Leptin could upregulate SOCS3 [74] (the expression levels of SOCS3 are higher in obese patients than in non-obese subjects) [75], guiding a reduced type I IFN response (via JAK–STAT), among other functions in diverse immune cells.

Author Response

Dear Editor.

I am pleased to resubmit for publication the revised version of ijms-1676603 “ The immune response in Adipocytes and their susceptibility to infection: A possible relationship with infectobesity”.  I appreciated the constructive criticism from the associated editor and reviewers. We addressed every comment, and highlighted them in red color and with track changes tool.

Reviewer:

Abstract: please, change word diverse with another one

  1. A. We are very grateful with this comment. We changed the word diverse for numerous, then the new phrase is: In addition, there is numerous evidence indicating that, during obesity, the immune system is ex-acerbated, suggesting an increased susceptibility of the patient to the development of several forms of illness or death

Introduction: adipose tissue description: it is not made only by white, brown and beige, but also pink adipocytes, please add also reference about it (for example, see the paper, Corrêa LH, Heyn GS, Magalhaes KG. The Impact of the Adipose Organ Plasticity on Inflammation and Cancer Progression. Cells. 2019 Jun 30;8(7):662. doi: 10.3390/cells8070662). 

Avoid to repeat references in this section (7,8 for example).

  1. A. We appreciate this comment, we added the word pink and adding the references: Corrêa LH, Heyn GS, Magalhaes KG. The Impact of the Adipose Organ Plasticity on Inflammation and Cancer Progression. Cells. 2019 Jun 30;8(7):662. doi: 10.3390/cells8070662) and Cinti, S. Pink Adipocytes. Trends Endocrinol Metab 2018, 29, 651-666, doi:10.1016/j.tem.2018.05.007, then the new phrase is: Adipose tissue consists of lipid-rich cells called adipocytes and can be divided into white, pink, beige and brown adipose tissue.

In fact, we added a brief explanation of these adipocytes in the section: 2.1 Adipose tissue’s composition in the second paragraph:

Adipose tissue is organized into discrete depots throughout the body [17,18]. Brown and beige adipocytes represent a small proportion of the total adipose tissue [18], with white adipose tissue (WAT) being the most abundant form, found in almost every area of the body, in fact, white adipocytes change their phenotype during pregnancy and lactation (transdifferentiation) developing pink adipocytes [19,20].

We corrected the references in this section.

Review section

The Role of Adipose Tissue in the Immune Response,

Adipose tissue can be classified by morphology into white, brown or beige subsets

The authors already described it. Please, delete it previously or here.

  1. We agreed and appreciated this comment, so, we delete this phare, then the new phrase started with: Adipose tissue is organized into discrete depots throughout the body [17,18].

Gut microbiota and its relationship with obesity,

At the beginning, please, describe better gut microbiota (it is not simply a natural component in our organism, as described), that is a complex microbial ecosystem that included other than bacteria, also virus, fungi and protozoa and parasites that play a fundamental role in immune system modulation and in maintaining good health for the host.

  1. We appreciated this comment, we included a better description of the microbial ecosystem:

In recent years, a natural component in our organism (known as the microbiota) has begun to be considered as a principal regulator in different physiological processes. The microbiota is a community  of microorganisms within a particular niche  whereas referring to the consortium  of all different  microorganisms and their genomes is known as the microbiome  [161,162]. Although bacteria have been identified within the main microorganisms that constitute this ecosystem, the participation of other biological entities such as ecosystem (bacteriophages) has begun to be considered [163-165], recent studies revealed that these non-bacterial microbial populations are also dynamic communities, interacting with one another and playing a vital role in host wellbeing [165]. The gut microbiota, which has a wide distribution in our body, with a weight of 1-2 kg. In addition, the genetic information in this biological community is 10 times greater than the human genome [166]. Among the main functions of the gut microbiota is protecting and maintaining the intestinal mucosa, through a symbiotic relationship with the host; additionally, plays important  roles, such as, supporting the immune development, and providing protective, structural, and metabolic functions essential for the human body [165]. This intimate relationship allows microorganisms to contribute to physiological homeostasis, and abnormalities in the microbiota participate in the development of various diseases [167]. In this sense, it has been proven that there is an essential connection between metabolic disorders, including obesity, and the gut microbiota, by showing that there is a regulation of fat storage, increasing energy collection, and regulating the formation of substrates for lipid storage [166,168,169].  

Please, correct the word adipocyte in the figure3 B.

  1. Thank you for your observation, We corrected the name in the figure 3b.

Acronym, please, specify, when it is possible The cJun-N-terminal-kinase (JNK), PTP1b, PPAR-γ, Mtb, and all the others present in this section, the first time that you use these terms.

On the other hand IFNs is twice defined. Please, correct it.

Please, replace term cues, it appears not clear.

  1. We corrected all the acronyms and putting the correct name of each one: JNK, PTP1b, PPAR-y, Mtb, TNF-a, IL1b, PI3K, MAPK, AMPK, ROS, IFN, NFkB, JAK-STAT, NOS, ERK, IRS1, mTOR, SOCS3, IRF3, GLUT4, FABP4, among others.

We corrected the definition of IFN.

We changed the term cues by either signals or factors depending of the context.

These sentence is not clear displayed in this way: Leptin could upregulate SOCS3 [74] (the expression levels of SOCS3 are higher in obese patients than in non-obese subjects) [75], guiding a reduced type I IFN response (via JAK–STAT), among other functions in diverse immune cells.

  1. We appreciated this comment, we modified the sentence for: Leptin could upregulate suppressor of cytokine signaling 3 (SOCS3) [74] (the expression levels of SOCS3 are higher in obese patients than in non-obese subjects) [75], this changes in SOCS3 expression impairs type I IFN response (via JAK–STAT), among other functions in diverse immune cells.

I look forward to your reply.

Sincerely,

Dr. León-Juárez Moisés

Departamento de Inmunobioquímica

Instituto Nacional de Perinatología ‘Isidro Espinosa de los Reyes’

Montes Urales #800, Col. Lomas de Virreyes

CP 11000, Ciudad de México, México.